# Current Trends in Endoscopic Diagnosis and Treatment of Early Esophageal Cancer

**DOI:** 10.3390/cancers13040752

**Published:** 2021-02-11

**Authors:** Franz Ludwig Dumoulin, Ralf Hildenbrand, Tsuneo Oyama, Ingo Steinbrück

**Affiliations:** 1Department of Medicine and Gastroenterology, Gemeinschaftskrankenhaus Bonn, Academic Teaching Hospital, University of Bonn, D-53113 Bonn, Germany; 2Institute for Pathology Bonn-Duisdorf, D-53123 Bonn, Germany; hildenbrand@patho-bonn.de; 3Department of Endoscopy, Saku Central Hospital Advanced Care Center, Saku, 3400-28 Nakagomie, Nagano, Japan; oyama@coral.ocn.ne.jp; 4Department of Medicine, Evangelisches Diakoniekrankenhaus Freiburg, Academic Teaching Hospital, University of Freiburg, D-79110 Freiburg, Germany; ingo.steinbrueck@diak-fr.de

**Keywords:** squamous cell esophageal cancer, gastro-esophageal reflux disease, Barrett’s esophagus, early adenocarcinoma of esophagus, endoscopic submucosal dissection, endoscopic mucosal resection

## Abstract

**Simple Summary:**

Early esophageal cancer is diagnosed in the context of reflux disease, surveillance of Barrett’s metaplasia, or during upper gastrointestinal endoscopy for other indications. High definition and virtual or dye chromoendoscopy are mandatory for the screening and evaluation of neoplasia. Endoscopic treatment options include endoscopic mucosal resection (EMR) or endoscopic submucosal dissection (ESD). Resection is considered curative if histopathology confirms low or absent risk of lymph node metastasis. Barrett’s high-grade dysplasia or early adenocarcinoma is treated by EMR or ESD, followed by ablation of Barrett’s epithelium to avoid metachronous cancer. ESD is the treatment of choice for squamous cell neoplasia. Excellent outcomes have been reported if the ESD of squamous cell cancer with slight submucosal infiltration and thus substantial risk for lymph node metastasis was combined with adjuvant chemo-radiotherapy. In contrast, infiltration of squamous cell cancer exceeding the lamina propria mucosae is not curative. However, despite a substantial risk of lymph node metastasis, excellent outcomes have recently been reported if endoscopic resection of tumors with up to 200 µm submucosal infiltration was combined with adjuvant chemo-radiotherapy.

**Abstract:**

Diagnosis of esophageal adenocarcinoma mostly occurs in the context of reflux disease or surveillance of Barrett’s metaplasia. Optimal detection rates are obtained with high definition and virtual or dye chromoendoscopy. Smaller lesions can be treated with endoscopic mucosal resection. Endoscopic submucosal dissection (ESD) is an option for larger lesions. Endoscopic resection is considered curative (i.e., without significant risk of lymph node metastasis) if histopathology confirms en bloc and R0 resection of a well-differentiated (G1/2) tumor without infiltration of lymphatic or blood vessels and the maximal submucosal infiltration depth is 500µm. Ablation of remaining Barrett’s metaplasia is important, to reduce the risk of metachronous cancer. Esophageal squamous cell cancer is associated with different risk factors, and most of the detected lesions are diagnosed during upper gastrointestinal endoscopy for other indications. Virtual high definition and dye chromoendoscopy with Lugol’s solution are used for screening and evaluation. ESD is the preferred resection technique. The criteria for curative resection are similar to Barrett’s cancer, but the maximum infiltration depth must not exceed lamina propria mucosae. Although a submucosal infiltration depth of up to 200 µm carries a substantial risk of lymph node metastasis, ESD combined with adjuvant chemo-radiotherapy gives excellent results. The complication rates of endoscopic resection are low, and the functional outcomes are favorable compared to surgery.

## 1. Introduction

According to a recent update from the Globoscan database, esophageal cancer is the seventh most prevalent cancer and the sixth most common cause of cancer-related mortality worldwide [1]. Over 90% of esophageal cancers are due to subtypes of squamous cell cancer or adenocarcinoma. Cancer-related mortality correlates with tumor stage at diagnosis with an overall five year survival rate of less than 20% [2]. While squamous cell cancer is the most prevalent subtype worldwide, dominant in Asia and Africa, there is an increasing prevalence of adenocarcinoma in high-income countries where this subtype today is more prevalent than squamous cell cancer [1,3,4,5]. This difference is thought to reflect the different distribution of risk factors, i.e., smoking, alcohol consumption, and environmental factors associated with squamous cell cancer versus obesity, gastro-esophageal reflux disease, and Barrett’s metaplasia as risk factors for adenocarcinoma. With the advent of high-definition endoscopy and the development of advanced endoscopic resection and ablation techniques, the concept of organ-preserving treatment for early esophageal cancer has been established with excellent survival rates and minimal associated morbidity. This review focuses on the latest trends in the endoscopic diagnosis and treatment of early esophageal neoplasia.

## 2. Barrett’s Esophagus, High-Grade Dysplasia, and Early Adenocarcinoma

### 2.1. Screening for the Presence of Barrett’s Esophagus

The incidences of gastro-esophageal reflux disease, Barrett’s esophagus, and esophageal adenocarcinoma are increasing in Western countries. Thus, screening for Barrett’s esophagus and surveillance for possible progression to high-grade dysplasia and (early) esophageal adenocarcinoma, including the option of endoscopic treatment, is a desirable concept. However, to date, there is no recommendation for population-based endoscopic screening. Instead, most guidelines suggest selected screening for long standing gastro-esophageal reflux disease with additional risk factors, such as age >50 years, male gender, Caucasian ethnicity, obesity, or family history of Barrett’s esophagus or adenocarcinoma [6,7,8,9]. A recent evaluation of these guidelines demonstrated poor test characteristics with either low or absent specificity leading to unnecessary endoscopies or unacceptably low sensitivity [10]. To overcome these problems, new technologies are being evaluated. Thus, promising data have been demonstrated for a non-endoscopic cytosponge test to detect the expression of the metaplasia biomarker trefoil factor 3. The method involves swallowing an encapsulated brush attached to a string. The capsule then dissolves in the stomach, and the expanded sponge is withdrawn to obtain the brush cytology [11]. Moreover, a breath test using an artificial intelligence supported sensor system to evaluate patterns of volatile organic compounds was shown to predict the presence of Barrett’s esophagus with high sensitivity and specificity [12]. Yet, with current guideline recommendations and pending further optimization of alternative non-endoscopic tests, the detection of Barrett’s esophagus still occurs most often in the setting of upper endoscopy performed for other indications than dedicated screening [5]. It is very unfortunate that a recent retrospective study on 123,395 upper gastrointestinal endoscopies calculated a miss rate for esophageal cancer of 6.5% with an associated two year survival rate of only 20% [13]. Thus, it seems to be more than justified to define a neoplasia detection rate as a quality indicator for upper endoscopy in patients with reflux disease [14].

### 2.2. Detection and Evaluation of High-Grade Dysplasia and Early Adenocarcinoma

Considerable efforts have been made to optimize the detection of dysplasia or early adenocarcinoma during surveillance endoscopy [15]. Important issues are the use of high-definition endoscopes, chromoendoscopy with acetic acid [16,17] and/or virtual chromoendoscopy [18,19], a sufficient inspection time (at least one minute per cm of segment length), and the application of a biopsy protocol with targeted biopsies from any suspicious lesion and four quadrant biopsies every 1–2 cm Barrett’s length (“Seattle protocol”) [9]. Several classification systems for the detection and evaluation of high-grade dysplasia and early adenocarcinoma in Barrett’s esophagus have been proposed. The Barrett’s International NBI Group (BING) classification [18] and the Japanese classification [19] both rely on surface and/or vessel irregularities identified with virtual narrow band imaging chromoendoscopy and show good sensitivity (80–87%) and specificity (88–97%) for the detection of dysplasia. The more recent proposed classification relies on surface irregularities and the loss of whitening of the mucosa after the application of acetic acid [16]. Using these descriptive criteria, a high sensitivity and a high negative predictive power for the presence or absence of Barrett’s neoplasia could be demonstrated not only for expert, but also for non-expert endoscopists (Table 1; Figure 1 a–d).

Acetic acid staining can also be helpful to evaluate the extent of Barrett’s cancer underneath the squamous epithelium [20]. In contrast to endoscopic image analysis (in particular with magnifying endoscopy), endoscopic ultrasound is less accurate in the prediction of infiltration depth [21]. However, endoscopic ultrasound is the most reliable method to detect possible lymph node metastasis and thus can be helpful in selected cases [22]. The additional acquisition of cytology with a specifically designed brush may increase the dysplasia detection rate and is included in the recommendations of the American Society for Gastrointestinal Endoscopy (ASGE) [9,23]. In the near future, artificial intelligence systems will become available to support the detection of dysplasia and early adenocarcinoma [24,25,26].

### 2.3. Endoscopic Treatment of Dysplasia and Early Adenocarcinoma

Pioneering work from Germany has established the concept of endoscopic mucosal resection (EMR) for high-grade dysplasia and mucosal esophageal adenocarcinoma [28]. This study included endoscopic resections for 1000 consecutive patients with T1a Barrett’s mucosal adenocarcinoma. During a follow up period of almost five years, the complete response rate was 96.3%. Surgery was necessary in 3.7% of the patients; metachronous neoplasia was detected in 14.5% and could be re-treated endoscopically in 81.4%, yielding a long-term complete remission rate of 93.8%. The same group also reported a series of 61 patients with endoscopic resection of low-risk T1b Barrett’s cancer: 90% of those with lesions ≤2 cm were in remission during a follow up of approximately four years; one patient developed a lymph node metastasis; and there was no tumor-related mortality [29]. Therefore, EMR is the preferred resection technique in Western countries [30] (Figure 2a–d). In contrast, endoscopic submucosal dissection (ESD) is the recommended method in the Japanese guidelines [21]. In fact, ESD is more appropriate for larger lesions, in particular Paris Type 0-Is, and is recommended for suspicious lesions for which en bloc resection is not possible using endoscopic mucosal resection [8,31]. Moreover, ESD compares favorably to EMR with higher en bloc (96% vs. 50%) and R0 resection rates (82% vs. 40%) with less recurrences (2.5% vs. 12.4%) [21]. Complication rates of endoscopic resection are low for both techniques [21,32]. They include perforation, bleeding, and, after resection of >70–80% of the circumference, also strictures. In cases of early adenocarcinoma, the criteria for curative resection are (i) complete (R0) resection without involvement of lateral and vertical margins, (ii) no invasion of lymphatic or blood vessels, and (iii) tumor grading G1/2. Infiltration depth into the submucosal layer of <500 µm is acceptable, in particular for patients with high perioperative risk [33]. While involved vertical margins are usually an indication for additional surgery, endoscopic controls and secondary resection are recommended in cases of positive lateral margins (Table 2).

### 2.4. Additional Treatment and Follow-Up

The development of metachronous neoplasia from residual Barrett’s metaplasia after endoscopic resection of high-grade dysplasia/early adenocarcinoma has been reported in up to 37% of cases during a two year follow up [34]. Thus, current guidelines recommend complete ablation of the remaining Barrett’s epithelium [31,33,35] (Figure 3). This is usually performed by radiofrequency ablation starting from ca. 5–10 mm above the squamo-columnar junction to 5–10 mm distal to the “neo Z-line”. Tissue ablation can also be achieved by argon plasma coagulation, cryo-ablation, or photodynamic therapy [35,36]. A major complication of ablation therapy is a stricture rate of approximately 5% [33]. A recently published study on 807 patients reported a 4.5% neoplasia recurrence rate after mucosal ablation, which peaked within the first 18 months [37].

Mucosal ablation can also be considered for confirmed non-visible low-grade dysplasia [38] or high-grade dysplasia and early cancer. However, a recent meta-analysis suggests that visible high-grade dysplasia or early cancer is best treated by a combination of endoscopic resection and additive ablation rather than using just radiofrequency ablation [39]. An endoscopic follow up after 3, 6, and 12 months and yearly thereafter has been suggested after endoscopic resection and/or mucosal ablation [33]. The approach to recurrent or metachronous neoplasia is similar to initial therapy [30] and the vast majority of these lesions can be re-treated endoscopically [28].

## 3. Squamous Cell High-Grade Dysplasia and Early Cancer

### 3.1. Screening for Squamous Cell Dysplasia or Early Cancer

As for Barrett’s esophagus, there is no established population-based screening for esophageal squamous cell cancer. However, endoscopic screening should be considered in the presence of risk factors, e.g., after diagnosis of head and neck cancer, in cases of long standing achalasia, or for persons with heavy smoking and drinking [40]. Vice versa, there is an increased risk for head and neck cancer in patients treated for superficial esophageal squamous cell cancer [41]. Screening for esophageal cancer is probably best performed by chromoendoscopy, after staining with a low concentration of Lugol’s solution or by virtual chromoendoscopy such as narrow band imaging [42,43]. A disadvantage of dye chromoendoscopy with Lugol’s solution, in particular at higher concentrations, is the induction of a painful inflammatory reaction. This inflammatory reaction interferes with the delineation of the borders of the lesion for up to several weeks. Thus, dye staining is usually done immediately before the beginning of endoscopic resection.

### 3.2. Detection and Evaluation of Squamous Cell Dysplasia or Early Cancer

Endoscopic evaluation of a detected lesion to predict the infiltration depth is critical since lymph node metastasis in squamous cell cancer is mainly associated with invasion depth [19]. The simplified classification of the Japan Esophageal Society uses vessel irregularities (loop formation, loss of loop formation, dilated and tortuous vessel) observed with regular or magnification endoscopy to predict infiltration depth [44] (Table 3; Figure 4a–d). The diagnostic accuracy is over 90% for B1 and B3 vessel patterns in predicting either superficial infiltration depth, i.e., epithelial (pT1a-EP) or lamina propria (pT1a-LPM) infiltration, or for the prediction of deep submucosal infiltration (pT1b-pSM2). In contrast, the vessel pattern B2 has an accuracy of only 55.7% for predicting an infiltration depth to the muscularis mucosae (pT1b-MM) or to the upper submucosal layer (pT1b-SM1). Endoscopic ultrasound has a relatively lower diagnostic accuracy for the evaluation of squamous cell cancer infiltration depth, but has its value as the most accurate diagnostic tool to assess possible lymph node metastasis [21,22].

### 3.3. Endoscopic Treatment

The recommended technique for endoscopic en bloc resection of early esophageal squamous cell cancer is ESD. Compared to EMR, ESD achieves higher en bloc (96% vs. 50%) and R0 resection rates (82% vs. 40%) and lower recurrence rates (2.5% vs. 12.4%) [21]. To facilitate the delineation of the lateral tumor extension, chromoendoscopy with Lugol’s solution is performed immediately before the procedure. The use of a traction device, e.g., the clip line technique [45], is useful to facilitate the resection in the narrow lumen of the esophagus. According to the recent Japanese guideline, endoscopic resection should be undertaken for all lesions with the B1 vessel pattern (predicted infiltration depth T1a-EP/LPM), unless the lesion is completely circumferential and has an axial extension of more than 5 cm. In addition, ESD should also be attempted for lesions with B2 vessel patterns (predicted infiltration depth T1a-MM/T1b-SM1) unless completely circumferential since there is a high probability that histopathology may reveal a more superficial infiltration depth and might thus be a curative resection (Figure 5 a–d). Curability is then evaluated by final histopathology: resection is considered curative for R0 resected superficial tumors without infiltration of lymphatic or blood vessels, i.e., pT1a-EP/LPM ly (-), v (-) (Table 4). In cases of vessel infiltration or infiltration of the submucosal layer, additional treatment (either surgery or chemo-radiotherapy) is recommended [21]. Data from observational studies comparing endoscopic resection with surgery for cT1a cancers demonstrate lower complication rates and associated health care costs with similar clinical outcomes for endoscopic treatment [21,46].

### 3.4. Additional Treatment and Follow-Up

Additional surgery or definitive chemo-radiotherapy is recommended after non curative resection [21] (Figure 6). Recently, excellent data have been published for adjuvant chemo-radiotherapy after complete endoscopic resection of T1b (SM1–2) cancers [47]. In this prospective observational study, 176 patients were treated with endoscopic resection of early esophageal squamous cell cancer. After histopathology, eighty-seven patients had either pT1a tumors with lympho-vascular invasion or pT1b tumors. These patients were treated by adjuvant chemo-radiotherapy (41.4 Gy) and had an overall three year survival of 90.7%. This outcome is comparable to published outcome data for surgery, and the rate of local recurrences is less than after primary chemo-radiotherapy [47]. The annual risk for metachronous esophageal is between 2.2 and 9.0%, and most recurrences can be treated endoscopically [19]. Thus, endoscopic follow-up and surveillance are recommended to be done in at least yearly intervals after curative local treatment. In addition, screening for synchronous head and neck, as well as lung cancers is also recommended [21].

## 4. Conclusions

Endoscopic detection and treatment of high-grade dysplasia and early esophageal cancer are established concepts for neoplasia with low or absent lymph node metastasis risk. Areas of research are (i) indications for endoscopic screening, (ii) optimizing of screening efficiency, and (iii) extending the concept of organ-preserving therapy by adjuvant treatment strategies. Screening may become more efficient with non-endoscopic pretesting, with promising data from cytosponge procedures or the use of an artificial intelligence-based breath test. The optimization of the neoplasia detection rate is important. Here, the widespread availability of high-definition endoscopy with virtual and/or dye-based chromoendoscopy is helpful. Moreover, additional support will soon be available from artificial intelligence image analysis systems. Finally, adjuvant treatment, e.g., chemo-radiotherapy may help to extend the curative ability of minimally invasive endoscopic therapy of early esophageal cancer in cases of high lymph node metastasis risk.

## Figures and Tables

**Figure 1 cancers-13-00752-f001:**
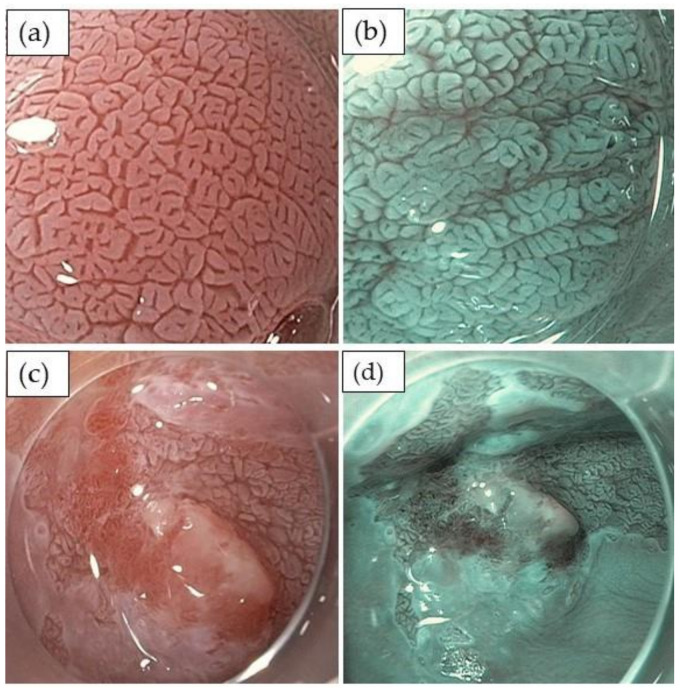
Endoscopic evaluation of Barrett’s esophagus and dysplasia with chromoendoscopy using 1.5% acetic acid. Top: non-dysplastic Barrett’s with a regular surface structure: (**a**) white light imaging, (**b**) narrow band imaging. Bottom: Barrett’s adenocarcinoma with the irregular surface structure and loss of acetowhitening: (**c**) white light imaging, (**d**) narrow band imaging.

**Figure 2 cancers-13-00752-f002:**
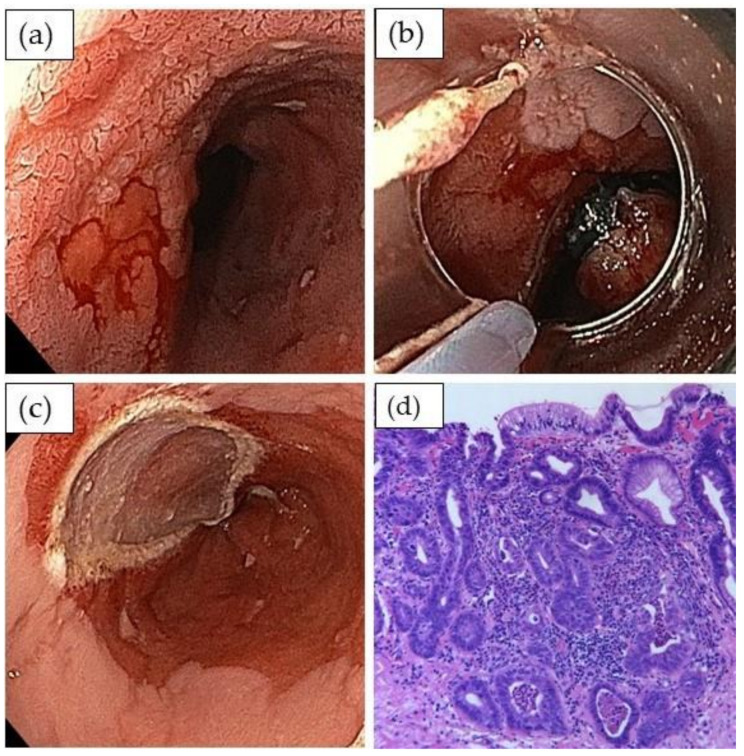
Endoscopic mucosal resection (EMR) of early adenocarcinoma with the suck and cut technique. (**a**) Long segment Barrett’s cancer with suspicious lesion to the left (white light imaging, acetic acid 1.5%); (**b**) lesion is band ligated and resected with the snare placed underneath the rubber band; (**c**) resection bed without any associated bleeding; (**d**) histopathology shows well differentiated intramucosal adenocarcinoma without infiltration of lymphatic or blood vessels pT1a m3, ly(-), v(-) G2 (H&E stain, × 400).

**Figure 3 cancers-13-00752-f003:**
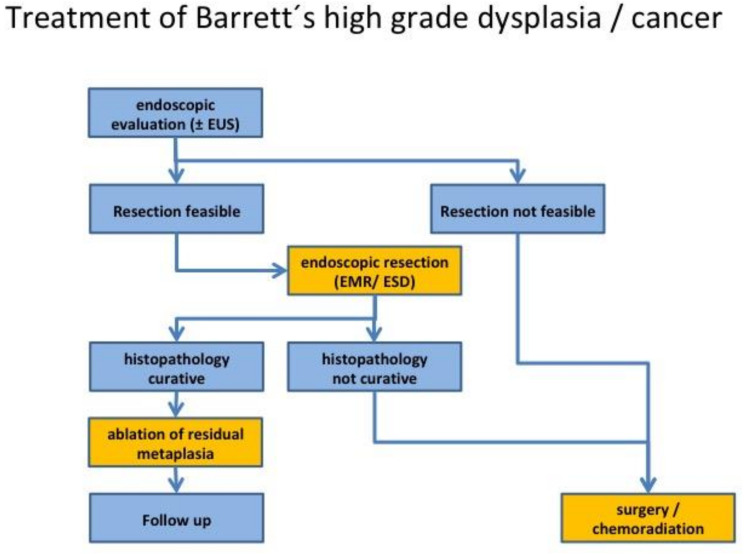
Proposed treatment algorithm according to current guidelines [30]. Evaluation may include endoscopic ultrasound to assess possible lymph node metastasis. Criteria for curative resection are listed in Table 2. EUS, endoscopic ultrasound; EMR, endoscopic mucosal resection; ESD, endoscopic submucosal dissection.

**Figure 4 cancers-13-00752-f004:**
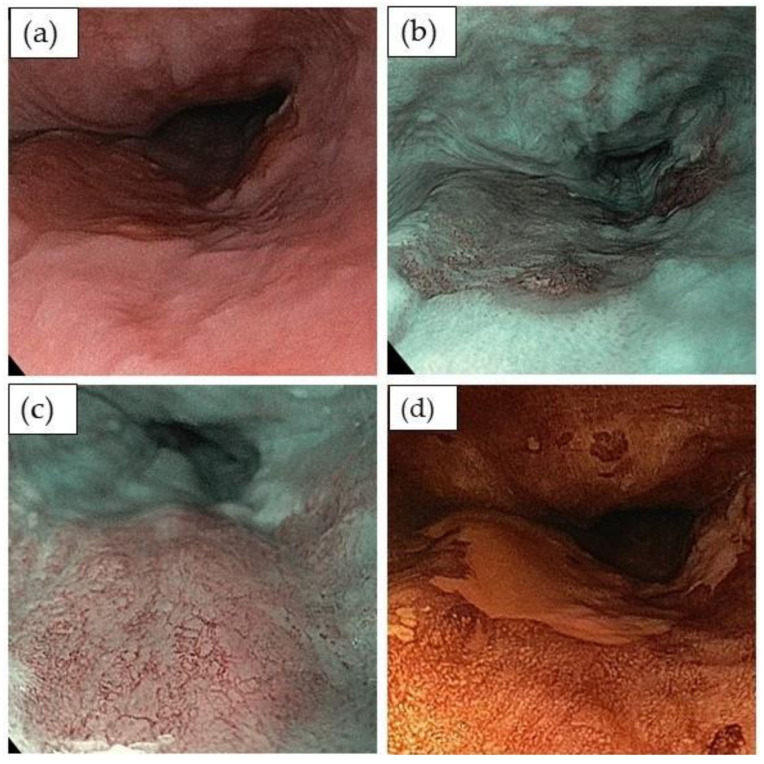
Endoscopic evaluation of early squamous cell cancer by endoscopic submucosal dissection: (**a**) irregular surface structure in the mid esophagus (white light imaging); (**b**) virtual chromoendoscopy of this area shows a brownish are with better definition of the borders (narrow band imaging); (**c**) close view of the lesion showing vessel irregularities (type B2 vessels); (**d**) chromoendoscopy with 0.5% Lugol’s solution nicely delineates the borders of the lesions and is ideally used immediately before starting an endoscopic resection.

**Figure 5 cancers-13-00752-f005:**
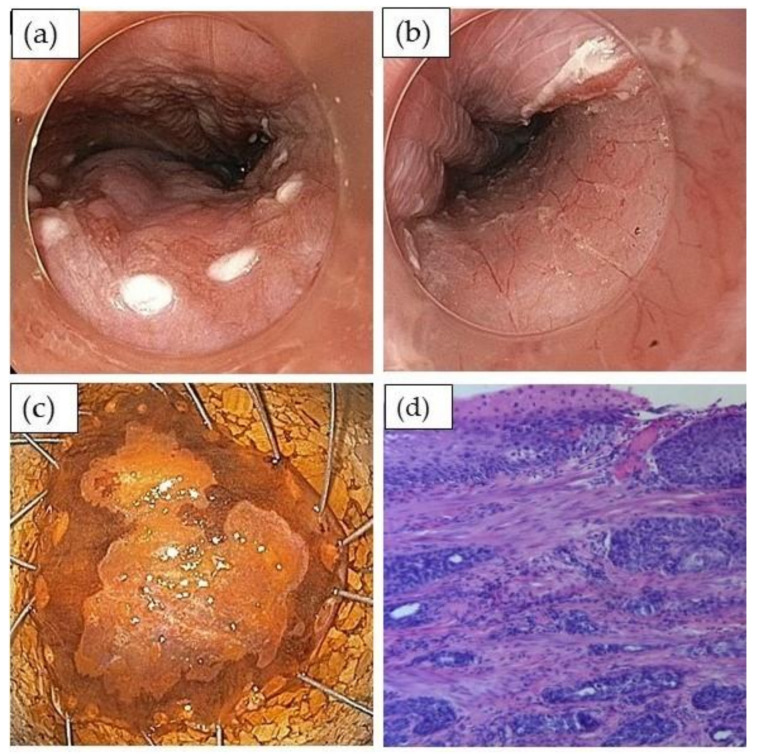
Endoscopic resection of early squamous cell cancer by endoscopic submucosal dissection: (**a**) after chromoendoscopy with Lugol’s solution, the borders of the lesion are marked with coagulation points; (**b**) resection bed without any associated bleeding; (**c**) specimen on corkboard (after repeated staining with Lugol’s solution); (**d**) histopathology shows poorly differentiated pT1b squamous cell cancer (H&E stain, × 400).

**Figure 6 cancers-13-00752-f006:**
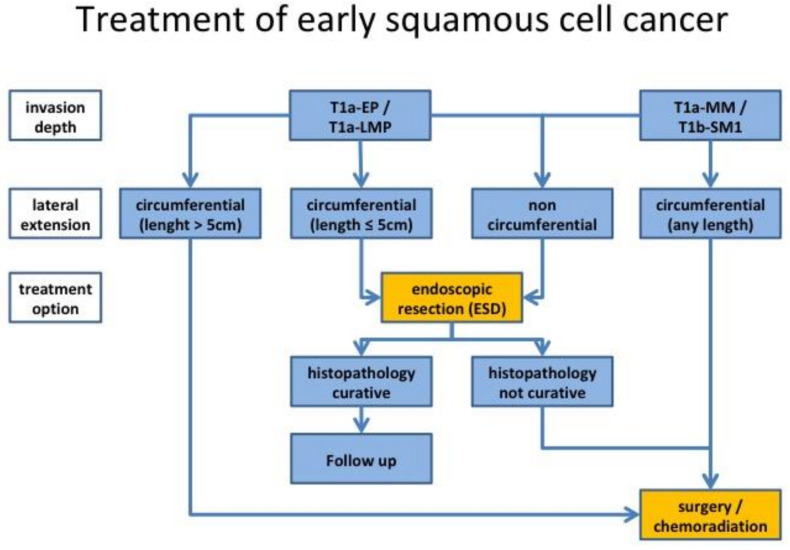
Proposed treatment algorithm of early esophageal squamous cell cancer according to current Japanese guidelines [21]. Evaluation should include endoscopic ultrasound to assess possible lymph node metastasis in selected cases. Invasion depth is diagnosed according to the Japan Esophageal Society (JES) classification (Table 3). Lateral extension is usually visualized by chromoendoscopy with Lugol’s solution. Criteria for curative resection are listed in Table 4. ESD, endoscopic submucosal dissection; EP, epithelial layer; LMP, lamina propria mucosae; MM, muscularis mucosae.

**Table 1 cancers-13-00752-t001:** Classification systems for Barrett’s associated neoplasia [16,18,19,27].

Classification System (Barrett’s Esophagus)
Paris Classification (‘generic” classification of superficial GI neoplasia)
0-Ip (pedunculated), 0-Is (sessile)
0-IIa (slightly elevated), 0-IIb (completely flat) 0-IIc (slightly depressed)
0-III (excavated/ulcerous)
Barrett’s International NBI Group (BING) Classification
mucosal pattern regular
• circular, ridged/villous, or tubular patterns
mucosal pattern irregular:
• absent or irregular patterns
vascular pattern regular
• blood vessels situated regularly along or between mucosal ridges and/or showing normal, long, branching patterns
vascular pattern irregular
• focally or diffusely distributed vessels not following the normal architecture of the mucosa
Japan Esophageal Society classification of Barrett’s esophagus
mucosal pattern regular:
• form/size: similar; arrangement: regular; density: low or same as surrounding area; white zone: clearly visible and/or of homogeneous width.
mucosal pattern irregular:
• form/size: variable; arrangement: irregular; density: high; white zone: obscure/invisible or of heterogeneous width.
vascular pattern regular:
• form: similar or bending and branching gently or regularly; caliber change gradual; location between or in mucosal pattern
vascular pattern irregular:
• form: various or bending and branching steeply or irregularly; caliber change: abrupt; location: beyond or regardless of mucosal patterns
flat pattern (classified as regular)
• completely flat surface without a clear demarcation line; greenish thick and/or long branching vessels
Portsmouth acetic acid (PREDICT) Classification
acetowhitening non-neoplastic:
• no focal loss of acetowhitening
acetowhitening neoplastic:
• focal loss of acetowhitening
surface pattern non-neoplastic:
• uniform evenly spaced pits with normal pit density
surface pattern neoplastic:
• compactly packed small pits with increased pit density; focal irregularity or disorganized pits; absent pit pattern

**Table 2 cancers-13-00752-t002:** Criteria for curative endoscopic resection of Barrett’s cancer [30].

Criteria for Curative Endoscopic Resection
• Resection en bloc/R0 (vertical margin)
• Grading G1/G21
• No infiltration of lymphatic/blood vessels
• Submucosal infiltration depth ≤500 µm

**Table 3 cancers-13-00752-t003:** Japanese classification for early squamous cell cancer [44].

Japan Esophageal Society (JES) Classification of Early Squamous Cell Cancer to Assess Tumor Infiltration Depth
Vascular pattern regular
Type A vessels
Type B vessels
Abnormal microvessels (severe irregularity/highly dilated abnormal vessels) • Type B1 with a loop-like formation • Type B2 without a loop-like formation • Type B3 highly dilated vessels, the calibers of which appear to be more than three times that of usual B2 vessels

**Table 4 cancers-13-00752-t004:** Criteria for curative endoscopic resection of early squamous cell cancer.

Criteria for Curative Endoscopic Resection
• Resection en bloc/R0
• Grading G1/G21
• No infiltration of lymphatic/blood vessels
• Infiltration depth pT1a-EP ^1^ or pT1a-LPM ^2^

^1^ Abbreviations: EP: epithelial layer; LPM: lamina propria mucosae; MM: muscularis mucosae; sm: submucosal. ^2^ The Japanese guideline recommends comprehensive evaluation of the need for additional treatment for pT1a-MM cancers, while additional treatment with surgery or chemoradiotherapy is strongly recommended for pT1b sm1 (≤200 µm) cancers [21].

## Data Availability

No new data were created or analyzed in this study. Data sharing is not applicable to this article.

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
