# Peer review of "Current Trends in Endoscopic Diagnosis and Treatment of Early Esophageal Cancer"

_cancers, 2021, doi:10.3390/cancers13040752_

Round 1

Reviewer 1 Report

This is a nicely written comprehensive review article. It covers the most information of recent guidelines for the management of early esophageal cancer by endoscopy.

Comments:

1. Although EUS is not good enough to distinguish T1 and T2, it has an important role to detect lymph node metastasis, even better than PET. Since lymph node metastasis is a contraindication for endoscopic therapy in patients with early esophageal cancer, the role of EUS should be well discussed.

2. Suggest authors to provide a table to compare the differences between early esophageal squamous cell carcinoma and adenocarcinoma if possible.

3. Suggest authors to provide the algorithm as when to and how to consider endoscopic therapy for patients with early esophageal cancer.

4. Do authors have any comments or suggestions for recurrent esophageal neoplasm after first time endoscopic treatment?

5.To improve readability, attention to English grammar and writing style is required throughout the manuscript.

Author Response

Many thanks for the helpful comments which we tried to incorporate as much as possible.

  • We have discussed the role of EUS in more detail (2.2., paragraph 2 and Figure 3; 3.2. paragraph 1).
  • We have elaborated the differences between early adenocarcinoma and squamous cell carcinoma in greater detail. In particular, we have included tables with the different criteria for curative endoscopic resection (2 / 4) and treatment algorithms (figure 3 / 6) for both entities.
  • We have given treatment algorithms for Barrett´s cancer and squamous cell cancer (fig. 3 / 6)
  • The aspect of recurrent / metachronous lesions is discussed for both AC and SCC including data on recurrence rates (2.4. first paragraph; 3.4. first paragraph).
  • We have tried to improve readability by correcting English grammar and writing style.

Reviewer 2 Report

Well written piece in a rapidly evolving topic. Two areas that can be improved I think that most readers will expect from this title are as follows:

  1. Classifications of lesions , both Paris, and JES (squam), should be included in one of two tables.
  2. The paper does focus on trends, but a Table of the best reported series in large EMR or ESD studies for both adenocarcinoma and SCC would add to the publication, wrt recurrence rates, progression to surgery, and cure rates based on intention to treat or intervention.

Author Response

Many thanks for the helpful comments which we tried to incorporate as much as possible.

  • We have now included a table with the classification systems for Barrett´s - Paris, BING, JES, PREDICT (table 1) and for SCC (table 2)
  • We have given more details from the landmark studies (2.3.; 3.3).

Reviewer 3 Report

The authors reviewed the latest trends in the endoscopic diagnosis and endoscopic treatment or comprehensive treatment of esophageal squamous cell cancer or adenocarcinoma. This review is well organized about them. There are comments below.

  1. In figure1, they showed the endoscopic figures with acetic acid 1.5%, and the legend of Figure1-d was “irregular vessels. However, this figure without magnification is not suitable for evaluating the vessel pattern. Generally, acetic acid makes it clear to evaluate the change of surface pattern. This legend should be revised, and the authors should describe about the efficacy of the acetic acid for the diagnosis of adenocarcinoma more in detail.
  2. The diagnosis of the extent of Barrett's esophageal cancer under the squamous epithelium is usually difficult. The authors should add this content.

Author Response

Many thanks for the helpful comments which we tried to incorporate as much as possible.

(1) We have revised the legend to figure 1 and have described acetic acid staining in more detail (2.2. – paragraph 1/2). We have also included classification systems (additional table #1).

(2) We have addressed the problem of cancer growth underneath squamous epithelium (2.2. - paragraph 2; new references #16/#20).

Round 2

Reviewer 1 Report

I am satisfied with the response by the authors to my comments and the addition they have made to the manuscript. This manuscript will be a good review of the endoscopic diagnosis and treatment of early esophageal cancer.
I still found a typing error in line 324, e"E"ndoscopic, please correct it.